# Effect of Different Dietary Lipid Sources on Growth Performance, Nutrient Digestibility, and Intestinal Health in Weaned Pigs

**DOI:** 10.3390/ani13193006

**Published:** 2023-09-24

**Authors:** Wenjuan Yang, Fei Jiang, Bing Yu, Zhiqing Huang, Yuheng Luo, Aimin Wu, Ping Zheng, Xiangbing Mao, Jie Yu, Junqiu Luo, Hui Yan, Jun He

**Affiliations:** 1Institute of Animal Nutrition, Sichuan Agricultural University, Chengdu 611130, China; muyao0462@163.com (W.Y.); ybingtian@yahoo.com.cn (B.Y.); zqhuang@sicau.edu.cn (Z.H.); luoluo212@126.com (Y.L.); wuaimin0608@163.com (A.W.); zpind05@163.com (P.Z.); acatmxb2003@163.com (X.M.); jerryyujie@163.com (J.Y.); junqluo2018@tom.com (J.L.); yan.hui@sicau.edu.cn (H.Y.); 2Key Laboratory of Animal Disease-Resistant Nutrition, Chengdu 611130, China; 3Singao Agribusiness Development Co., Ltd., Longyan 361000, China; jiangfeifeijiang@163.com

**Keywords:** lipid source, growth performance, intestinal health, weaned pigs

## Abstract

**Simple Summary:**

Lipids are an ideal source of energy for piglets. Several studies have shown that dietary oil supplementation can improve the growth performance of weaned piglets. This experiment investigated the effects of different fat sources on the growth performance and intestinal health of weaned piglets, and compared with the conventional use of soybean oil, pigs treated with fish–palm–rice oil mixture (FPRO) and coconut–palm–rice oil mixture (CPRO) showed improvements in digestibility and intestinal epithelial function, and a reduction in the production of inflammatory cytokines. The results of this study can provide the theoretical basis for the rational selection of lipid sources in weaned piglets and other mammals.

**Abstract:**

To investigate the effects of lipid sources on growth performance and intestinal health, 72 weaned pigs were randomly allocated to three treatments. Pigs were fed with a corn–soybean meal diet containing 2% soybean oil (SO), or fish–palm–rice oil mixture (FPRO), or coconut–palm–rice oil mixture (CPRO). The trial lasted for 28 days; blood and intestinal tissue samples were collected. The results showed that the crude fat digestibility of the FPRO group was higher than that of the SO and CPRO groups (*p* < 0.05). The FPRO group also had higher digestibility of dry matter, ash, and gross energy than the SO group (*p* < 0.05); compared to the SO group, the serum interlukin-6 (IL-6) concentration was decreased. Interestingly, the FPRO and CPRO groups had higher villus height than the SO group in the jejunum and ileum, respectively (*p* < 0.05). Moreover, the FPRO group had higher *Lactobacillus* abundance than the SO group in the colon and cecum (*p* < 0.05). Importantly, the expression levels of tight junction protein ZO-1, Claudin-1, and Occludin in the duodenal and ileal mucosa were higher in the FPRO group than in the SO and CPRO groups (*p* < 0.05). The expression levels of nutrient transporters such as the CAT-1, PepT1, FATP1, and SGLT1 were higher in the FPRO group than in the SO group (*p* < 0.05). The improved digestibility and intestinal epithelium functions, as well as the reduced inflammatory cytokines, in the FPRO and CPRO group suggest that a mixed lipid source such as the FPRO deserves further attention.

## 1. Introduction

The intestinal mucosa consists of a simple columnar layer of epithelial cells and the lamina propria and muscular mucosa beneath it, with epithelial cells on the mucosal surface creating a barrier between the sometimes hostile external and internal environments [1,2]. The largest exchange surface between the body and the external environment is the intestinal epithelial barrier (IEB), whose permeability plays a central role in nutrient intake and body fluid regulation and controlling the infection of pathogens [3]. However, various factors, such as weaning stress, oxidative stress, and bacterial infection, can impair the IEB by inducing overproduction of pro-inflammatory cytokines and reactive oxygen species (ROS), which subsequently leads to decreases in nutrient digestion and absorption, systemic inflammatory response, and even the death of piglets [4,5,6,7].

Lipids can serve as an ideal energy source for piglets due to its high energy concentration (2.25 times more than carbohydrates and proteins), and have been usually added to the diet to solve the problem of insufficient energy intake during the weaning stage of piglets [8]. A number of studies indicated that dietary oil supplementation can improve the growth performance of weaned piglets. For instance, dietary lipid supplementation significantly increased ADG and G:F of piglets from the second week to the fifth week post-weaning, and significantly increased ADG and G:F of the piglets during the first five weeks post-weaning [9]. Moreover, dietary lard supplementation significantly increased the G:F of piglets from the second week to the third week and the third week post-weaning [10].

In addition to acting as an energy source, the lipids can also provide a series of essential fatty acids (EFAs), such as linoleic acid and linolenic acid. Previous studies indicated that the EFAs can inhibit the overproduction of intestinal inflammatory mediators, especially various pro-inflammatory cytokines [11,12]. For instance, in vitro studies indicated that Eicosapentaenoic acid (EPA) and Docosahexaenoic acid (DHA) can inhibit the production of IL-1β and tumor necrosis factor (TNF)-α by monocytes [13], and inhibit the production of IL-6 and IL-8 by venous endothelial cells [14,15]. Moreover, dietary supplementation of 2% Conjugated linoleic acid (CLA) in second-trimester sows significantly reduced intestinal inflammation, and increased the serum concentrations of immunoglobulin in neonatal pigs upon enterotoxigenic *Escherichia coli* challenge [16].

A previous study indicated that the effect of dietary lipids supplementation on animals may vary widely depending on their sources or chemical structures [17]. For instance, dietary supplementation of fish oil (one of the major sources of EPA) significantly reduced the production of inflammatory cytokines such as the TNF-α, IL-1β, and IL-6 by macrophages in rodents [18,19]. Rice oil contains a large number of compounds with antioxidant properties, such as glutamine, tocopherol, and tocotrienol [20]. Previous studies have shown that dietary supplementation of rice oil can improve the growth performance of broilers [21,22] and reduce the serum cholesterol concentration [23]. Compared to long-chain fatty acids (LCFAs), medium-chain fatty acids (MCFAs) are more efficiently digested and absorbed in the small intestine of piglets [24]. Coconut oil is a critical source of the MCFAs, and a previous study indicated that the ADG of the weaned pigs during the first four weeks post-weaning was higher in the coconut oil group than in the butter and corn oil group (the weaner diet contains 8% oil) [25]. Moreover, both the ADG and ADFI of the weaned pigs were higher in the coconut oil group than in the soybean oil group during the three to four weeks after weaning [9].

Although, these studies indicated a beneficial effect of different lipid sources on the weaned pigs, the combined effects of different lipid sources are just beginning to be explored. In the present study, we investigated the effects of different lipid sources on growth performance and intestinal health in weaned piglets. Pigs were fed with the corn–soybean meal diet containing 2% SO or oil mixtures (FPRO and CPRO). We observed improved digestibility and intestinal epithelium functions, as well as a reduced production of inflammatory cytokines in the FPRO and CPRO group, which may offer a theoretical basis for rational selection of lipid sources for the weaned pigs as well as other mammalian animals.

## 2. Materials and Methods

### 2.1. Experimental Design and Diets

A total of 72 healthy weaned pigs (average 7.80 ± 0.11 kg) with similar body conditions and breeds were randomly divided into three treatment groups with eight replicates per treatment, and three piglets housed in a pen. Pigs were fed corn- and soybean-meal-based diets containing 2% soybean oil (SO), fish–palm–rice oil mixture (FPRO, consisting of 10% fish oil, 50% palm oil, and 40% rice oil), or coconut–palm–rice oil mixture (CPRO, consisting of 5% coconut oil, 80% palm oil, and 15% rice oil), respectively. The composition and nutrition level of the basic diet was prepared in reference to the NRC (2012) pig feeding standard (Table 1) [26]. The basal diet was supplemented with 2% soybean oil and the FPRO and CPRO groups were supplemented with 2% FPRO and CPRO, respectively. The feeding management and immunization of the experimental group were carried out according to the field regulations. The pre-feeding period lasted for 3 days. The experiment lasted 28 days. Pigs were supplied with feed and fresh water freely and room temperature was controlled between 25~28 °C, relative humidity 65% ± 5% [27]. On days 0 and 29 of the experimental period, the feed volume and weight of piglets were recorded to calculate the average daily feed volume (ADFI), average daily gain (ADG), and feed conversion rate (FCR) for the entire experimental period.

### 2.2. Sample Collection

Feed samples were collected at the beginning of the experiment and stored at −20 °C for nutritional analysis. Fresh fecal samples were collected immediately after excretion on days 25–28 of the experiment, weighed, and 10 mL 10% H_2_SO_4_ solution was added to every 100 g of fresh fecal matter [28]. Feed and fecal samples were dried for 2 days at 65 °C, ground through a one-mesh sieve, and then stored at −20 °C until nutrient digestibility analysis. On the morning of the 29th day, we selected one piglet from each cage that is closest to the average weight for blood collection and slaughter, and collected 15 mL of venous blood. After 30 min, the blood was centrifuged at 3500 rpm and 4 °C for 15 min to prepare the serum. After centrifugation, the serum was packed separately (1.5 mL EP tube, 300 μL/tube) and put it on ice. All blood samples were repackaged and then transferred to −20 °C refrigerator for storage. After slaughter, about 2 cm of the middle segment of the duodenum, jejunum, and ileum were taken and rinsed gently with precooled normal saline to remove the contents, and were then immersed in 4% paraformaldehyde for fixation and preservation, which was used to determine the histomorphology [29]. Next, another segment of the duodenum, jejunum, and ileum was cut open with a scissor, then the contents of the intestine were washed with normal saline and the water was absorbed with filter paper. The intestinal mucosa samples were gently scraped and immediately frozen with liquid nitrogen, and then stored at −80 °C. The cecum and colon chyme were squeezed into a paper cup and quickly collected with the corresponding sampling spoon. The chyme was put into the corresponding cryopreservation tube (sterilized) [30]. Each kind of intestine was put into four tubes, wrapped with tin foil, and quickly put into liquid nitrogen for preservation.

### 2.3. Apparent Total Tract Nutrient Digestibility Analysis

Using Acid insoluble ash (AIA) as an endogenous indicator, the nutrient digestibility levels of dried, ground, and fecal samples were analyzed. The contents of dry matter (DM), crude protein (CP), crude fat (EE), and ash were determined according to AOAC [31]. The energy (GE) was measured directly by a bomb calorimeter.

### 2.4. Serum Biochemical Indicators

Immunoglobulin (IgA, IgG, IgM), glycolipid substitute (triglyceride, cholesterol, low-density lipoprotein, blood glucose, etc.), inflammatory factor (IL-6, TNF-α, IL-1β), antioxidant indexes (T-AOC, CAT, SOD, GSH-PX, MDA), hormone and growth factors (insulin, IGF-1) were tested in special kits built in Nanjing. The specific operation was carried out according to the kit instructions.

### 2.5. Intestinal Morphology

The intestinal segments fixed with 4% paraformaldehyde were dehydrated through a graded series of ethanol and then embedded in paraffin [32]. The slices (2 µm) were stained with hematoxylin and eosin, observed with Leica DM1000 biological microscope under 100 times of field of vision, selected 10 appropriate fields of vision (complete villi) and photographed, and then measured the villus height and corresponding crypt depth of a single villus [33].

### 2.6. Intestinal Disaccharidase Content

The frozen duodenum, jejunum, and ileal mucosa were homogenized with a ratio of 1:9 (*w*/*v*) frozen normal saline for 15 min. Homogenates were centrifuged at 3500× *g* at 4 °C for 15 min, and enzyme activities were measured in the supernatant. The content of disaccharidase (sucrase, lactase, and maltase) in the supernatant was determined with the kit of Nanjing Jiancheng Bioengineering Research Institute.

### 2.7. Intestinal Flora

We weighed 0.2 g caecum and colon thawed chyme, and used EZNA^®^ Stool DNA kit (Omega Bio Tek, Norcross, GA, USA) to extract total DNA in accordance with the specific operation per the instructions of the kit. The Taqman probe was used for real-time fluorescence quantitative PCR reaction, and RealMasterMix (Probe) was used for detection. The number of copies contained in each sample was calculated by the Ct value and the standard curve with each gram of content as the detection unit. The result was expressed by the common logarithm of the number of bacteria in each gram of content (lg (copies/g)). Construction of the 20 μL reaction system: 8 μL 2 × RealMasterMix, 1 μL of forward and 1 μL of reverse primers (100 nM), 1 μL 20 × Probe Enhancer Solution, 0.3 μL probe, 1 μL DNA, and 7.7 μL of RNase-Free ddH_2_O. Reaction procedure: pre denaturation at 95 °C for 10 s, denaturation at 95 °C for 15 s, annealing at 60 °C for 25 s, extension at 72 °C for 60 s, 40 cycles.

### 2.8. Gene Expression

About 0.1 g of the intestinal sample was put into a pre-cooled 1.5 mL centrifuge tube, and 1 mL RNAiso Plus reagent (Takara, Kyoto, Japan) was added. The samples were homogenized on ice until dissolved. After standing for 5 min, the samples were centrifuged for 15 min at 12,000× *g*/4 °C. An amount of 400 μL supernatant was absorbed into a new aseptic centrifuge tube, and isopropyl alcohol was added in the same volume, inverted, and mixed gently. After standing on ice for 10 min, the samples were centrifuged at 12,000× *g*/4 °C for 10 min to precipitate RNA. The supernatant was discarded, and 1 mL of pre-cooled 75% ethanol (freshly mixed with DEPC water) was added, then the suspended RNA was precipitated and centrifuged for 5 min at 7500× *g*/4 °C. Discard the supernatant and leave it at room temperature for 1–2 min to dry the precipitate. When the RNA was slightly dry, 40 μL DEPC water was added, and the pipette gun was used to gently blow and suck 50 times to fully dissolve the RNA. By using a spectrophotometer (NanoDrop 2000, Thermo Fisher Scientific, Inc., Waltham, MA, USA) to detect the purity and concentration of total RNA, samples with an OD260/OD280 ratio of 1.8–2.0 were considered suitable. Total RNA was used as template to synthesize cDNA by reverse transcription kit, referring to the manual of PrimeScript™ RT Reagent (Takara, Japan) for specific operation. Reverse transcription consists of two steps: 42 °C, 2 min (removal of genomic DNA); 37 °C for 15 min, 85 °C for 5 s. Based on the procedure and reaction system of the TB Green™ Premix Ex Taq™ II (TaKaRa, Japan) kit, QuanStudio™ 6 Flex real-time quantitative PCR was used to detect the expression of target genes in the small intestine. The reaction system of qPCRas follows: TB Green Premix Ex Taq II (Tli RNaseH Plus, 2×) 5 μL, ROX ReferenceDye II (50×) 0.2 μL, forward primer 0.4 μL, reverse primer 0.4 μL, cDNA 1 μL, Double Steaming Water 3 μL. The reaction procedure as follows: 95 °C predenaturation for 30 s, 95 °C denaturation for 5 s, 60 °C annealing for 34 s, 40 cycles. Quantitative results used the GAPDH gene as the internal reference gene. The relative expression levels of each target gene in tissues were calculated by2^−ΔΔCt^ [34].

### 2.9. Data Statistics and Analysis

After all experimental data were analyzed by Microsoft Excel 2019, SPSS 24.0 statistical software was used to perform one-way ANOVA on the experimental data of the three treatment groups. When the differences were significant, Duncan’s multiple comparisons were performed. All experimental data were expressed as “mean ± standard error”, *p* < 0.05 was considered significant, and 0.05 ≤ *p* ≤ 0.10 was considered a trend.

## 3. Result

### 3.1. Effect of Different Dietary Lipid Sources on Growth Performance and Nutrient Digestibility in Weaned Pigs

Compared to the SO group, dietary supplementation of FPRO and CPRO had no influence on ADFI and ADG (Table 2). The apparent digestibility of DM, EE, GE, and ash was higher in the FPRO group than in the SO group (*p* < 0.05) (Table 3). Moreover, the apparent digestibility of ash was also higher in the CPRO group than in the SO group (*p* < 0.05).

### 3.2. Effects of Different Dietary Lipid Sources on Serum Immunoglobulin, Pro-Inflammatory Factors, and Antioxidant Capacity

As shown in Table 4, the serum IgA concentration in the FPRO group was higher than that in the SO group (*p* < 0.05). In addition, the serum concentrations of pro-inflammatory factors such as IL-6 and IL-1β in the FPRO and CPRO groups were lower than those in the SO group (*p* < 0.05). Compared to the SO group, the serum CAT capacity was higher in the FPRO and CPRO groups (*p* < 0.05).

### 3.3. Effects of Different Dietary Lipid Sources on Intestinal Morphology and Mucosal Enzyme Activities

As shown in Table 5 and Figure 1, the FPRO and CPRO groups had a higher villus height than the SO group in the jejunum and ileum, respectively (*p* < 0.05). The crypt depth of the FPRO and CPRO group was also lower than that of the SO group (*p* < 0.05). From Table 6, it can be seen that the maltase activity of the duodenal and jejunal mucosa in the FPRO and CPRO groups is higher than in the SO group (*p* < 0.05). Compared to the SO group, the lactase activity tended to be higher in the FPRO and CPRO group (0.05 < *p* < 0.10).

As shown in Table 7, the FPRO group had a higher abundance of *Lactobacillus* than the SO group in the colon and cecum (*p* < 0.05). Compared to the SO group, the CPRO group had a higher abundance of *Lactobacillus* in the cecum and a higher abundance of *Bifidobacterium* in the colon, respectively (*p* < 0.05).

### 3.4. Effects of Different Dietary Lipid Sources on Expression Levels of Key Genes Related to Intestinal Epithelial Function and Mucosal Inflammatory Factors

As shown in Figure 2, the expression levels of tight junction protein ZO-1, Claudin-1, and Occludin in the duodenal and ileal mucosa were significantly higher than that of the SO and CPRO groups (*p* < 0.05). The expression levels of nutrient transporters such as the CAT-1, PepT1, FATP1, and SGLT1 were higher in the FPRO group than in the SO group (*p* < 0.05). At the same time, compared to the SO group, the expression level of IL-1β in jejunum and ileum mucosa in FPRO and CPRO groups was significantly decreased (*p* < 0.05). The expression level of ERK1 in duodenum of FPRO group was also significantly decreased (*p* < 0.05), and the expression level of NF-κB in ileum mucosa of the CPRO group was also significantly decreased (*p* < 0.05).

## 4. Discussion

Fat and oil are commonly considered as effective energy sources for animals, including pigs [35]. The digestion and metabolism of lipids are dependent on their sources, dietary proportions, and intermolecular distribution of saturated and unsaturated fatty acids [36,37]. Previous studies on the weaned pigs (5 to 20 kg) indicated that dietary fat supplementation reduced feed intake, but significantly improved the feed conversion efficiency [8]. Compared to other vegetable oils, palm oil and coconut oil are mainly composed of MCFA (the fatty acids below 16 carbon atoms in coconut oil account for 66.2% of the total fatty acids) [38], and their digestibility levels range from 80 to 95% [39]. Long-chain fatty acids can delay stomach emptying and increase feelings of fullness, which may decrease the feed intake for pigs [40]. In the present study, we discussed the effects of different lipid mixtures on the growth performance and intestinal health of weaned piglets. As expected, the digestibility levels of DM, GE, and ash were higher in the FPRO and CPRO group than in the SO. Moreover, the FPRO group had a higher digestibility of EE than other groups, which may be associated with the presence of a considerable amount of unsaturated fatty acids in the fish oil [41].

Immunoglobulins, also known as antibodies, are glycoprotein molecules produced by plasma cells, which act as a critical part of the immune response by specifically recognizing and binding to particular antigens, such as bacteria or viruses, and aiding in their destruction [42]. In this study, dietary supplementation of FPRO can significantly increase the serum IgA concentration, indicating an elevated immunity in the pigs. Pro-inflammatory cytokines, such as IL-1β and IL-6, can mediate host inflammatory processes and produce a rapid immune response after infection with pathogenic microorganisms [43]. In this study, the serum IL-1β and IL-6 concentrations were decreased in the FPRO and CPRO group. This is probably due to the fish and coconut oil in the two lipid mixtures, as previous studies indicated that dietary supplementation of coconut oil and fish oil can reduce the occurrence of inflammatory responses in the weaned pigs [44,45]. Interestingly, the serum activity of CAT was also higher in the FPRO and CPRO group than in the SO group. CAT, as a key enzyme in the antioxidant defense system, plays an important role in clearing hydrogen peroxide in the body and avoiding oxidative stress [46]. A previous study indicated that palm oil contains a large number of antioxidants, which can improve the activity of antioxidant enzymes [47], whereas rice oil contains a large number of compounds with antioxidant properties, such as glutamine, tocopherol, and tocotrienol [21].

The digestion and absorption of various nutrients are basically carried out in the small intestine. Villus height and crypt depth of intestinal mucosal epithelium are closely related to the absorption capacity of intestinal mucosa [48]. In this study, the villus height of the jejunum and ileum in the FPRO and CPRO groups was higher than in the SO group, respectively, and the crypt depth in FPRO and CPRO groups was also lower than in the SO group. Coconut oil and palm oil are rich in medium-chain fatty acids [49], MCFA can be directly utilized by intestinal epithelial cells for energy production, thereby helping to support the integrity of the piglets’ intestines [50]. Previous studies have shown that MCFA supplementation for the weaned pigs can improve the intestinal morphology and integrity, as indicated by the significant increase in intestinal villus length, and decrease in crypt depth and the number of intraepithelial lymphocytes [51]. Fish oil has also been reported to increase the villus height and villus height/crypt depth ratio in the weaned pigs [6]. Changes of intestinal morphology are usually accompanied by brush margins of intestinal epithelium digestive enzyme activity [52]. In this study, the maltase and lactase activities of duodenal and jejunal mucosa in FPRO and CPRO groups were higher than that in the SO group.

The gut microbiota is an important component of the gut ecosystem, and is closely related to the integrity of intestinal epithelial cells, the digestion and metabolism of nutrients, immune responses, and diseases [17,53]. Some difficult-to-digest fats can reach the hindgut and be a source of nutrients for bacteria, which may regulate the gut bacteria balance [54]. Different sources of oils have different compositions, so their effects on gut microbes are also different [55]. In this study, the abundance of lactobacillus in the colon and cecum in the FPRO group was higher than in the SO group. Moreover, the abundance of cecal lactobacillus i and colonic bifidobacterium in the CPRO group was also higher than in the SO group. This result is consistent with previous studies that found that dietary supplementation of rice and coconut oil significantly increased the abundances of beneficial bacteria such as *bifidobacterium* and *Lactobacillus* in the intestine [56,57].

Tight junctions play a crucial role in maintaining intestinal permeability, consisting of transmembrane barrier proteins (such as claudin and occludin), cytoplasmic scaffold proteins (such as the ZO family), and adhesion molecules [58]. A previous study indicated that fish oil can enhance the expression levels of occludin and claudin-1 in the weaned pigs [6]. In this study, the expression levels of ZO-1, Claudin-1, and Occludin in the duodenum and ileum were higher in the FPRO group than in the SO and CPRO group. Moreover, the expression levels of nutrient transporters such as the CAT-1, PepT1, FATP1, and SGLT1 were higher in the FPRO group than in the SO group. PepT1 is a small peptide transporter, while CAT1 is mainly responsible for the transport of basic amino acids [59,60]. GLUT-2 is one of the major transporters of glucose absorption, while FATP-1 is responsible for the transport of long-chain fatty acids into intestinal cells [61]. These results indicated improved epithelium functions in the weaned pigs upon FPRO supplementation. IL-1β is considered to be an important pro-inflammatory cytokine involved in innate immunity; in general, down-regulating the expression of IL-1β protein in intestinal tissue can reduce inflammatory damage [62]. NF-κB is a promoter of inflammation-associated tumors, and it is essential for promoting inflammation-associated cancers [63]. Previous studies indicated that dietary supplementation of coconut oil and fish oil can reduce the occurrence of inflammatory responses in the weaned pigs [44,45]. In this study, the expression levels of IL-1β in the jejunum and ileum were down-regulated in the FPRO and CPRO group. Moreover, the expression levels of duodenal ERK1 and ileal NF-κB were decreased in the FPRO and CPRO groups, respectively. These results may suggest an anti-inflammatory effect of a lipid mixture containing the fish and coconut oil.

## 5. Conclusions

In the present study, we explored the effects of different lipid sources on growth performance and intestinal health in weaned pigs. We observed improved digestibility and intestinal epithelium functions, as well as a reduced production of inflammatory cytokines in the FPRO and CPRO groups, which may offer a theoretical basis for rational selection of lipid sources for the weaned pigs as well as other mammalian animals. Of course, this study has a relatively short duration and may have certain limitations. Further experimental verification is needed regarding the effects of mixed oil FPRO and CPRO on fattening pigs and other animals.

## Figures and Tables

**Figure 1 animals-13-03006-f001:**
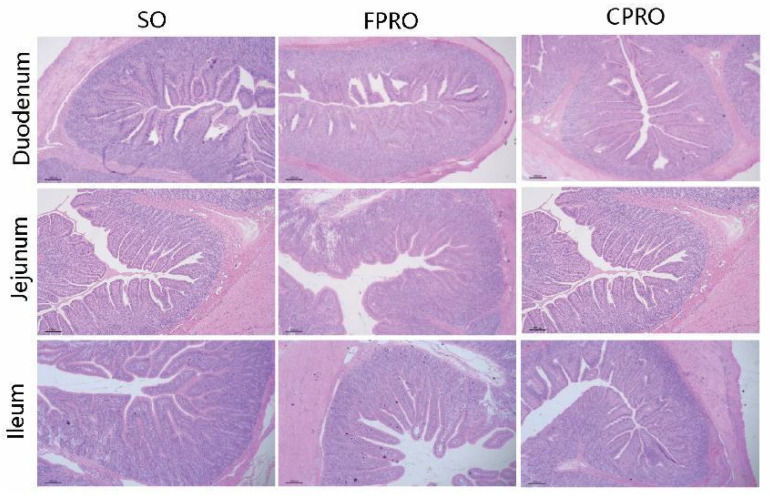
Effects of different dietary lipid sources on intestinal morphology (H&E; ×40).

**Figure 2 animals-13-03006-f002:**
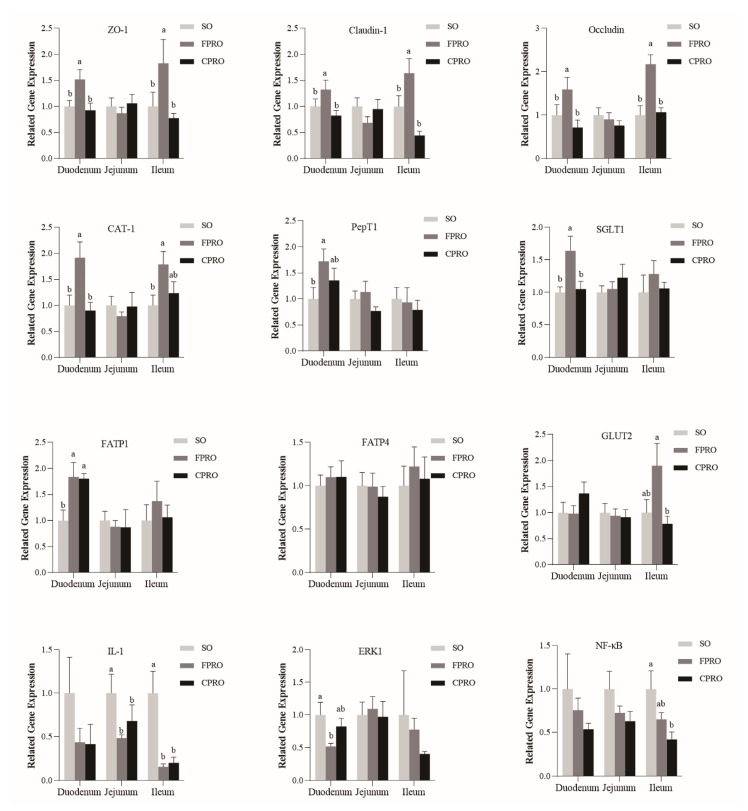
The expression levels of tight junction proteins. a, b mean values within a row with unlike superscript letters were significantly different (*p* < 0.05).

**Table 1 animals-13-03006-t001:** Formulated diet and nutrient compositions.

Ingredients	%	Nutrient Level	Contents
corn [Grade 8.7%]	39.56	Digestible energy of pigMC/kg	3.5
Hulled barley	8	Crude protein %	17.77
Broken rice	15	Calcium %	1.19
Wheat bran	3.5	Total phosphorus %	1.11
Soybean meal	22.6	Available phosphorus %	0.54
Secondary powder ^1^	5	digestible lysine %	1.36
Oil ^2^	2	digestible methionine %	0.32
Lysine	0.61	digestible cystine %	0.57
DL Methionine	0.08	digestible threonine %	0.77
L-threonine	0.25		
L-Tryptophan	0.1		
Choline chloride 50%	0.1		
Vitamin premix ^3^	0.05		
Mineral premix ^4^	0.25		
NaCl	0.3		
Limestone	0.9		
Calcium hydrogen phosphate	0.9		
Complex enzyme	0.05		
Pig phytase (5000)	0.05		
Compound acidifier	0.7		
Total	100		

^1^ Also known as black flour, yellow flour, lower- or third-class flour, it is a byproduct of wheat processing mainly composed of finely ground bran and some wheat endosperm. It is one of the byproducts obtained from grinding various types of flour using wheat grains as raw materials. ^2^ The oil added in the SO group is 2% soybean oil; the oil added in FPRO group is 2% mixed oil 1 (palm oil:rice oil:fish oil = 5:4:1); the oil added in CPRO group is 2% mixed oil 2 (palm oil:rice oil:coconut oil = 16:3:1). ^3^ The vitamin premix provided the following per kg of diet: 9000 IU of VA, 3000 IU of VD3, 20 IU of VE, 3 mg of VK3, 1.5 mg of VB1, 4 mg of VB2, 3 mg of VB6, 0.02 mg of VB12, 30 mg of niacin, 15 mg of pantothenic acid, 0.75 mg of folic acid, and 0.1 mg of biotin. ^4^ The mineral premix provided the following per kg of diet: 100 mg Fe, 6 mg Cu, 100 mg Zn, 4 mg Mn, 0.30 mg I, 0.3 mg Se.

**Table 2 animals-13-03006-t002:** Effects of different dietary lipids on growth performance of weaned piglets.

ITEM	Treatments			SEM	*p*-Value
	SO	FPRO	CPRO		
Initial weight (kg)	7.91	7.52	7.97	0.11	0.231
Final weight (kg)	18.63	18.11	18.84	0.34	0.668
ADFI (g/d)	611.43	604.52	633.15	11.92	0.686
ADG (g/d)	362.96	378.27	380.19	9.47	0.736
F:G	1.69	1.60	1.66	0.02	0.436

ADFI, average daily feed intake; ADG, average daily gain; F:G, feed:gain ratio. (1) Mean and total SEM are listed in separate columns (*n =* 8). (2) SO, pigs were fed with a basal diet; FPRO, pigs fed with 2% mixed oil 1; CPRO, pigs fed with 2% mixed oil 2.

**Table 3 animals-13-03006-t003:** Effects of different dietary lipids on nutrient digestibility of weaned piglets.

ITEM	Treatments			SEM	*p*-Value
	SO	FPRO	CPRO		
DM (%)	80.35 ^b^	84.99 ^a^	83.41 ^ab^	0.71	0.017
EE (%)	79.17 ^b^	84.22 ^a^	80.39 ^b^	0.62	0.001
GE (%)	80.93 ^b^	85.84 ^a^	83.15 ^ab^	0.76	0.007
ASH (%)	55.65 ^b^	60.31 ^a^	59.42 ^a^	0.71	0.020
CP (%)	78.43	79.82	77.10	0.84	0.422

DM, dry matter; EE, ether extract; CP, crude protein; GE, Gross Energy. (1) Mean and total SEM are list in separate columns (*n =* 8). (2) a, b mean values within a row with unlike superscript letters were significantly different (*p* < 0.05). (3) SO, pigs were fed with a basal diet; FPRO, pigs fed with 2% mixed oil 1; CPRO, pigs fed with 2% mixed oil 2.

**Table 4 animals-13-03006-t004:** Effects of different dietary lipids on serum indexes and antioxidant capacity of weanling piglets.

ITEM	Treatments			SEM	*p*-Value
	SO	FPRO	CPRO		
IgA (μg/mL)	26.85 ^b^	31.18 ^a^	29.42 ^ab^	0.70	0.032
IgG (μg/mL)	306.52	332.89	332.92	8.27	0.338
IgM (μg/mL)	13.65	14.71	14.43	0.24	0.203
IGF-1 (μg/L)	31.01	28.96	30.12	0.41	0.133
INS (mIU/L)	65.86	66.52	66.46	1.08	0.964
IL-6 (ng/L)	867.10 ^a^	762.34 ^b^	681.82 ^b^	24.24	0.016
IL-1β (ng/L)	33.12 ^a^	27.29 ^b^	27.00 ^b^	1.13	0.039
TNF-α (pg/mL)	277.83	243.12	273.70	8.28	0.155
Glucose (mmol/L)	5.47	4.86	5.82	0.18	0.108
Triglyceride (mol/L)	2.02	2.15	2.23	0.15	0.871
T-SOD (U/L)	62.69	65.58	65.99	1.14	0.472
AOC (U/mL)	2.89	3.23	2.79	0.33	0.861
MDA (nmol/mL)	3.88	4.20	4.45	0.21	0.547
GSH-PX (μmol/L)	674.96	1024.32	986.35	80.37	0.198
CAT (U/mL)	18.71 ^b^	39.80 ^a^	38.57 ^a^	3.49	0.014

(1) Mean and total SEM are list in separate columns (*n =* 8). (2) a, b mean values within a row with unlike superscript letters were significantly different (*p* < 0.05). (3) SO, pigs were fed with a basal diet; FPRO, psigs fed with 2% mixed oil 1; CPRO, pigs fed with 2% mixed oil 2.

**Table 5 animals-13-03006-t005:** Effects of different dietary lipids on intestinal morphology of weaned piglets.

ITEM	Treatments			SEM	*p*-Value
	SO	FPRO	CPRO		
Duodenum					
Villus height (μm)	562.85	573.50	547.88	11.41	0.687
Crypt depth (μm)	297.50	295.27	293.96	7.66	0.985
V:C	1.92	1.86	1.88	0.04	0.639
Jejunum					
Villus height (μm)	443.21 ^b^	463.16 ^b^	519.49 ^a^	11.36	0.009
Crypt depth (μm)	243.53 ^a^	201.61 ^b^	200.73 ^b^	7.10	0.021
V:C	2.18	2.13	2.02	0.038	0.418
Ileum					
Villus height (μm)	547.32 ^b^	637.04 ^a^	542.90 ^b^	16.43	0.018
Crypt depth (μm)	317.15	284.84	277.49	9.12	0.272
V:C	1.92	2.02	1.86	0.03	0.178

V:C, Villus height:Crypt depth. (1) Mean and total SEM are list in separate columns (*n =* 8). (2) a, b mean values within a row with unlike superscript letters were significantly different (*p* < 0.05). (3) SO, pigs were fed with a basal diet; FPRO, pigs fed with 2% mixed oil 1; CPRO, pigs fed with 2% mixed oil 2.

**Table 6 animals-13-03006-t006:** Effects of different dietary lipids on intestinal mucosal enzymes in weaned piglets.

ITEM	Treatments			SEM	*p*-Value
	SO	FPRO	CPRO		
Duodenum					
AKP (U/g prot)	15.45	14.92	12.19	0.80	0.341
Maltase (ng/L)	303.96 ^b^	370.63 ^a^	335.14 ^ab^	9.85	0.015
Lactase (pg/mL)	203.89	203.75	209.01	3.69	0.815
Sucrase (ng/L)	191.27	192.21	185.64	4.78	0.844
sIgA (μg/mL)	22.29	24.35	24.66	2.86	0.374
Jejunum					
AKP (U/g prot)	21.33 ^b^	24.78 ^a^	23.60 ^ab^	0.67	0.105
Maltase (ng/L)	120.63 ^b^	185.47 ^a^	187.18 ^a^	12.52	0.026
Lactase (pg/mL)	187.70	162.64	169.69	6.82	0.290
Sucrase (ng/L)	190.33	159.71	169.18	8.42	0.200
sIgA (μg/mL)	27.40	28.52	36.48	4.00	0.637
Ileum					
AKP (U/g prot)	17.56	16.68	17.47	0.61	0.844
Maltase (ng/L)	353.42	355.95	359.95	7.20	0.938
Lactase (pg/mL)	283.35	318.31	322.21	7.86	0.079
Sucrase (ng/L)	243.25	240.86	255.44	4.71	0.418
sIgA (μg/mL)	5.62	4.33	4.83	0.58	0.684

APK, alkaline phosphatase. (1) Mean and total SEM are list in separate columns (*n =* 8). (2) a, b mean values within a row with unlike superscript letters were significantly different (*p* < 0.05). (3) SO, pigs were fed with a basal diet; FPRO, pigs fed with 2% mixed oil 1; CPRO, pigs fed with 2% mixed oil 2.

**Table 7 animals-13-03006-t007:** Effects of different dietary lipids on intestinal microbial population in weaned piglets.

ITEM	Treatments			SEM	*p*-Value
	SO	FPRO	CPRO		
Cecum, log (copies/g)					
*Escherichia coli*	6.23	5.59	5.52	0.25	0.468
*Lactobacillus*	6.45 ^b^	7.29 ^a^	7.52 ^a^	0.17	0.018
*Bifidobacterium*	4.43 ^b^	4.74 ^ab^	5.89 ^a^	0.29	0.089
*Bacillus*	8.24	8.25	8.17	0.06	0.868
Colon, log (copies/g)					
*Escherichia coli*	5.45	5.40	5.18	0.20	0.859
*Lactobacillus*	6.82 ^b^	7.91 ^a^	7.33 ^ab^	0.19	0.070
*Bifidobacterium*	4.49 ^b^	4.63 ^b^	6.18 ^a^	0.25	0.025
*Bacillus*	8.49	8.39	8.31	0.05	0.440

(1) Mean and total SEM are list in separate columns (*n =* 8). (2) a, b mean values within a row with unlike superscript letters were significantly different (*p* < 0.05). (3) SO, pigs were fed with a basal diet; FPRO, pigs fed with 2% mixed oil 1; CPRO, pigs fed with 2% mixed oil 2.

## Data Availability

The data used to support the findings of this study are available from the corresponding author upon request.

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
