# Peer review of "Effect of Different Dietary Lipid Sources on Growth Performance, Nutrient Digestibility, and Intestinal Health in Weaned Pigs"

_animals, 2023, doi:10.3390/ani13193006_

Round 1
Reviewer 1 Report
Dear authors,
Thank you for the submission of your research article.
While research article is valid and clear, the execution of the manuscript requires some serious revisions if to be considered.
Overall, the materials and methods section of the report is unclear and in some instances quite vague. The duration of the study needs to be indicated for example, when were pigs weaned? from what age were diets provided? When were diets withdrawn?
What are the number of animals used for each measurement. As mentioned 72 pigs were used in the study however, in results table 2, 3,5 it is mentioned in the footnotes, "n=8". While this may be a sufficient samples size of 8 pigs per treatment of gut morphology etc. I think this is insufficient when considering performance results and should have monitored the performance of all 72 pigs on trial and may explain the lack of significance found in performance.
In terms of piglet performance there is no mention of how ADG was calculated as it does not mention piglet weights at any stage in the report.
Diet analysis results are not provided in the results section and the statistical analysis section is vague. Information of model structure is required such as main effects, if random factors were considered etc. Without most of this information it is difficult to assess the validity of the results discussed.
These are a few of a number of issues I have found when reviewing the manuscript which contribute to my overall recommendation to reject the manuscript at this time.
There are some minor spelling/typo errors throughout the manuscript along with double spacing after full stops or references for example, see line 65 "[11, 12]" . Line 77 "cdietary" should be "dietary". Please read through manuscript for these minor errors.
When numbers appear in the text e.g. line 82 "4 weeks", any numbers less than 10 should be written in long form so should read "four weeks". Digits can be used when the number is 10 or greater, or if number if followed by a unit such as % or kg.
Author Response
Dear editor and reviewers,
Thank you very much for giving us an opportunity to submit a revised version of our manuscript. We appreciate for your comments and suggestions concerning our manuscript. These comments are valuable and helpful for revising and improving our manuscript, and we revised the manuscript in accordance with the detailed comments and suggestions. All revisions are highlighted in red in the text. The point-by-point revisions to the comments and suggestions are listed as follows:
Overall, the materials and methods section of the report is unclear and in some instances quite vague. The duration of the study needs to be indicated for example, when were pigs weaned? from what age were diets provided? When were diets withdrawn?
Re: Thanks for your comments.I have stated in the materials and methods of the newly submitted manuscript that we are using 28 day weaned piglets, which will start the formal experiment after 3 days of pre feeding. The experiment will last for 28 days, and on the 29th day, we will weigh, collect blood, and slaughter them.
What are the number of animals used for each measurement. As mentioned 72 pigs were used in the study however, in results table 2, 3,5 it is mentioned in the footnotes, "n=8". While this may be a sufficient samples size of 8 pigs per treatment of gut morphology etc. I think this is insufficient when considering performance results and should have monitored the performance of all 72 pigs on trial and may explain the lack of significance found in performance.
Re: Thanks for your comments.During our experiment, we used 72 animals and divided them into 3 experimental groups, with 24 pigs in each group. We were divided into 8 replicates, each with 3 pigs. At the end of the experiment, one pig with the closest body weight to the average was selected from each of the three replicates for slaughter and sampling.
In terms of piglet performance there is no mention of how ADG was calculated as it does not mention piglet weights at any stage in the report.
Re: Thanks for your comments.Considering the issue of growth performance, I have added the initial and final weights of piglets in Table 2 of the newly submitted manuscript
Reviewer 2 Report
The authors reported the effects of different dietary lipid mixtures on growth performance, nutrient digestibility, and intestinal health in weaned pigs and the topic is interesting. There are some points that must be clarify. Methods used in this study must be written in meaningful way. In fact, it is not clear the choice of the fat in the fat blend and the amount of inclusion. Fatty acid composition of the experimental mixtures and experimental diet are not reported in the manuscript. The length of experimental trial and the sample number is missing. Moreover, also statistical evaluation must be rewrite for clarity.
Minor comments are reported below:
Line 14: write in full the acronyms.
Line 61: Why do the authors mention arachidonic acid? I suggest deleting it and include linolenic acid
Line 77: dietary. Delete c
Line 97: add the initial weight of the piglets and the weaning age.
Line 102: Why did the authors decide to test these fat blends? On what basis was the composition of the blends decided?
Line 105: The formulation, chemical composition and fatty acid composition of the three experimental diets must be reported.
Line 107: the length of the experimental study should be inserted.
Line 118- 139: the Sample collection and processing paragraph is not clear. Divide it in sampling and insert sample processing in the following paragraphs. Moreover, the sample numerosity for blood, fecal sample and intestinal tract samples is missing.
Line 129: were all the animals slaughtered? at what age? Please add information.
Line 142:” end of the experiment”. The length of the experimental study should be inserted.
Line 208: Data statistics and analysis. Please better specify the statistical method for all the variable considered, also adding the experimental unit.
Table 2. Add initial and final weight of the animals.
Line 286: Add the discussion section.
Line 368: Please rewrite the conclusion section, considering the result about the two oil mixtures.
Author Response
Point to point response to reviewer’s comments
Dear editor and reviewers,
Thank you very much for giving us an opportunity to submit a revised version of our manuscript. We appreciate for your comments and suggestions concerning our manuscript. These comments are valuable and helpful for revising and improving our manuscript, and we revised the manuscript in accordance with the detailed comments and suggestions. All revisions are highlighted in red in the text. The point-by-point revisions to the comments and suggestions are listed as follows:
Line 14: write in full the acronyms.
Re: Thanks for your comments. It has been corrected in the revised manuscript.
Line 61: Why do the authors mention arachidonic acid? I suggest deleting it and include linolenic acid
Re: Thanks for your comments. It has been corrected in the revised manuscript.
Line 77: dietary. Delete c
Re: Thanks for your comments. It has been corrected in the revised manuscript.
Line 97: add the initial weight of the piglets and the weaning age.
Re: Thanks for your comments. It has been corrected in the revised manuscript.
Line 102: Why did the authors decide to test these fat blends? On what basis was the composition of the blends decided?
Re:Thanks for your comments.When we choose the mixture of oils and fats, we use the ideal fatty acid model as a guide, and the performance of these oils and fats is better than that of a single oil, and the fatty acid structure is better balanced.
Line 105: The formulation, chemical composition and fatty acid composition of the three experimental diets must be reported.
Re:Thanks for your comments.This question is answered later in Table 1.
Line 107: the length of the experimental study should be inserted.
Re: Thanks for your comments. It has been corrected in the revised manuscript.
Line 118- 139: the Sample collection and processing paragraph is not clear. Divide it in sampling and insert sample processing in the following paragraphs. Moreover, the sample numerosity for blood, fecal sample and intestinal tract samples is missing.
Re: Thanks for your comments. It has been corrected in the revised manuscript.
Line 129: were all the animals slaughtered? at what age? Please add information.
Re: Thanks for your comments. It has been corrected in the revised manuscript.
Line 142:” end of the experiment”. The length of the experimental study should be inserted.
Re: Thanks for your comments. It has been corrected in the revised manuscript.
Line 208: Data statistics and analysis. Please better specify the statistical method for all the variable considered, also adding the experimental unit.
Re: Thanks for your comments. It has been corrected in the revised manuscript.
Table 2. Add initial and final weight of the animals.
Re: Thanks for your comments. It has been corrected in the revised manuscript.
Line 286: Add the discussion section.
Re: Thanks for your comments. It has been corrected in the revised manuscript.
Line 368: Please rewrite the conclusion section, considering the result about the two oil mixtures.
Re: Thanks for your comments. It has been corrected in the revised manuscript.

Reviewer 3 Report
The manuscript submitted by Yang et al. investigated the effect of 2 different integrations of lipid sources (in particular, fish and coconut oil in combination with palm and rice oil) compared with soybean oil in weaned pigs. The work is correctly structured, and the principal effects on growth performance, nutrient digestibility, and intestinal health were intensively investigated.
Main considerations:
How were the different ratios of fat sources (FPRO group: 10% fish oil, 50% palm oil, and 40% rice oil; CPRO group: 5% coconut oil, 80% palm oil, and 15% rice oil) evaluated in the different groups? Please explain: by previous studies, or balancing the chemical composition.
Please in Table 1 specify better what is stone powder (calcium carbonate?) and secondary powder (?)
In Table 1 please delete the word pig from "pig digestible", ...
In Table 2 I think it is better to separate growth performance from nutrient digestibility. In Table 2 the body weight data is missing.
In Table 2 there is no present full name of GE
For your statistical analysis of growth performances do you consider the pen as experimental unit? For microbiological and serological data we consider the pig as an experimental unit?
In the Materials and Methods section, the sequence of probe and primer used in this paper can be listed or indicated as a reference.
Line 178-179: Primer and probe used for intestinal flora investigations.
Line 203-204: forward and reverse primer, Line 206 housekeeping gene GAPDH (sequence primer)
Minor issues.
Line 62: Please consider substituting study with studies (the references are 2)
Line 77: Please correct cdietary with dietary
Line 101: Please correct plam with palm
Line 184: Please add RNAiso Plus reagent (Takara, Japan)
Line 191: Please consider substituting “for 1-2 min to dry and precipitate” with “for 1-2 min to dry the precipitate”.
Author Response
Dear editor and reviewers,
Thank you very much for giving us an opportunity to submit a revised version of our manuscript. We appreciate for your comments and suggestions concerning our manuscript. These comments are valuable and helpful for revising and improving our manuscript, and we revised the manuscript in accordance with the detailed comments and suggestions. All revisions are highlighted in red in the text. The point-by-point revisions to the comments and suggestions are listed as follows:
How were the different ratios of fat sources (FPRO group: 10% fish oil, 50% palm oil, and 40% rice oil; CPRO group: 5% coconut oil, 80% palm oil, and 15% rice oil) evaluated in the different groups? Please explain: by previous studies, or balancing the chemical composition.
Re:Thanks for your comments.When we choose the mixture of oils and fats, we use the ideal fatty acid model as a guide, and the performance of these oils and fats is better than that of a single oil, and the fatty acid structure is better balanced.
Please in Table 1 specify better what is stone powder (calcium carbonate?) and secondary powder (?)
Re: Thanks for your comments. It has been corrected in the revised manuscript.
In Table 1 please delete the word pig from "pig digestible", ...
Re: Thanks for your comments. It has been corrected in the revised manuscript.
In Table 2 I think it is better to separate growth performance from nutrient digestibility. In Table 2 the body weight data is missing.
Re: Thanks for your comments.Considering the large amount of data in the article, growth performance and nutrient digestibility have been combined, and weight data has been supplemented.
In Table 2 there is no present full name of GE
Re: Thanks for your comments. It has been corrected in the revised manuscript.
For your statistical analysis of growth performances do you consider the pen as experimental unit? For microbiological and serological data we consider the pig as an experimental unit?
Re: Thanks for your comments.We use columns as the unit for growth performance statistics, as we select one pig per column for blood samples and slaughter, so we use individual pigs as the unit.
Line 62: Please consider substituting study with studies (the references are 2)
Re: Thanks for your comments. It has been corrected in the revised manuscript.
Line 77: Please correct cdietary with dietary
Re: Thanks for your comments. It has been corrected in the revised manuscript.
Line 101: Please correct plam with palm
Re: Thanks for your comments. It has been corrected in the revised manuscript.
Line 184: Please add RNAiso Plus reagent (Takara, Japan)
Re: Thanks for your comments. It has been corrected in the revised manuscript.
Line 191: Please consider substituting “for 1-2 min to dry and precipitate” with “for 1-2 min to dry the precipitate”.
Re: Thanks for your comments. It has been corrected in the revised manuscript.

Reviewer 4 Report
The manuscript of Wenjuan Yang et al. presents interesting research about the use of alternative feed in pig feeding. Below some comments and suggestions to the authors.
The abstract is too long. According to the journal guidelines, the abstract should be about 250 words. The exposition can be improved.
A simple summary section is missing. I suggest reducing the abstract and creating the Simple Summary section. In this section, the authors can explain and address the problems faced in their study.
Line 55. Please report EFAs being for plural. Check this in the entire manuscript.
Line 58. EPA and DHA are not stated before abbreviating.
Line 60. Explain the abbreviation CLA.
In the introduction section the following sentence “Pigs were fed with the corn-soybean meal diet containing 2% SO or oil mixtures (FPRO and CPRO).” should be mentioned in the M&M section.
The following statement “We observed an improved digestibility and intestinal epithelium functions, as well as a reduced production of inflammatory cytokines in the FPRO and CPRO group, which may offer a theoretical basis for rational selection of lipid sources for the weaned pigs as well as other mammalian animals.” should be stated in the conclusion.
The exposition of the topic in the introduction can be improved.
In the experimental design section, authors should indicate the duration of the study. This information is only mentioned in the abstract.
Do not the authors believe the 28-day period is short? In my opinion, it should be a limitation of the study. The authors could create a short paragraph at the end of the manuscript explaining the limitation of the study and further perspectives. Alternatively, they could integrate this aspect in the conclusion.
Line 72. LCFA and MCFA should be reported in plural form then LCFAs and MCFAs. Check all abbreviations in the abbreviations section.
Line 136. If an Abbreviation section is included in the manuscript, please use abbreviations.
The letter P of the p-value should be reported as a lowercase letter.
Figure 1. Regarding the effects of different dietary lipid sources on intestinal morphology, it could be interesting if the Authors could illustrate an intestinal section of an individual who did not receive the dietary treatment to observe differences between a control subject and treated animals.
I appreciate whether the authors could include in the bibliography of their manuscript the following research, highlighting in the discussion section the relevance of the use of alternative feedstuffs in pigs' nutrition for improving animal health, and growth performance.
Sutera, A.M.; Arfuso, F.; Tardiolo, G.; Riggio, V.; Fazio, F.; Aiese Cigliano, R.; Paytuví, A.; Piccione, G.; Zumbo, A. Effect of a Co-Feed Liquid Whey-Integrated Diet on Crossbred Pigs’ Fecal Microbiota. Animals 2023, 13, 1750. https://doi.org/10.3390/ani13111750
In my opinion, the exposition of the manuscript can be improved.
Author Response
Dear editor and reviewers,
Thank you very much for giving us an opportunity to submit a revised version of our manuscript. We appreciate for your comments and suggestions concerning our manuscript. These comments are valuable and helpful for revising and improving our manuscript, and we revised the manuscript in accordance with the detailed comments and suggestions. All revisions are highlighted in red in the text. The point-by-point revisions to the comments and suggestions are listed as follows:
A simple summary section is missing. I suggest reducing the abstract and creating the Simple Summary section. In this section, the authors can explain and address the problems faced in their study.
Re: Thanks for your comments. It has been corrected in the revised manuscript.
Line 55. Please report EFAs being for plural. Check this in the entire manuscript.
Re: Thanks for your comments. It has been corrected in the revised manuscript.
Line 58. EPA and DHA are not stated before abbreviating.
Re: Thanks for your comments. It has been corrected in the revised manuscript.
Line 60. Explain the abbreviation CLA.
Re: Thanks for your comments. It has been corrected in the revised manuscript.
In the introduction section the following sentence “Pigs were fed with the corn-soybean meal diet containing 2% SO or oil mixtures (FPRO and CPRO).” should be mentioned in the M&M section.
Re: Thanks for your comments. It has been corrected in the revised manuscript.
The following statement “We observed an improved digestibility and intestinal epithelium functions, as well as a reduced production of inflammatory cytokines in the FPRO and CPRO group, which may offer a theoretical basis for rational selection of lipid sources for the weaned pigs as well as other mammalian animals.” should be stated in the conclusion.
Re: Thanks for your comments. It has been corrected in the revised manuscript.
In the experimental design section, authors should indicate the duration of the study. This information is only mentioned in the abstract.
Re: Thanks for your comments. It has been corrected in the revised manuscript.
Line 72. LCFA and MCFA should be reported in plural form then LCFAs and MCFAs. Check all abbreviations in the abbreviations section.
Re: Thanks for your comments. It has been corrected in the revised manuscript.
The letter P of the p-value should be reported as a lowercase letter.
Re: Thanks for your comments. It has been corrected in the revised manuscript.

Round 2
Reviewer 1 Report
It is clear that efforts have been made to improve the manuscript. The introduction reads well with only a few minor typos that can be amended in editing. The materials and methods are much clearer now, thank you to the authors for incorporating the changes suggested.
I suggest the title of table 1 should read "Table 1. Formulated diet and nutrient compositions". Where the diets analysed for composition? If so, please include a table of the analysed figures in the results section. If not, then please include a statement that diets were not analysed after formulation.
In 2.9 Data statistics and analysis section should it read "and 0.05 ≥ p ≤ 0.10 was considered a trend" instead?
Thank you to the reviewer for including initial and final weights of piglets, the results section reads well.
The discussion section has been clearly improved and reads well.
Thank you again to the authors for their efforts to improve the manuscript.
Improvements have been made to the manuscript to improve consistency in language used.
Some minor editing for adjustments of typos e.g. like 66 "(DHA )" - remove spacing and line 68 "[14,15].Moreover"- space required after full stop.
Table 2 footnotes "2the" T should be capatilzed
Reformatting of reference no. 13
Author Response
I suggest the title of table 1 should read "Table 1. Formulated diet and nutrient compositions". Where the diets analysed for composition? If so, please include a table of the analysed figures in the results section. If not, then please include a statement that diets were not analysed after formulation.
Re: Thanks for your comments. It has been corrected in the revised manuscript.
In 2.9 Data statistics and analysis section should it read "and 0.05 ≥ p ≤ 0.10 was considered a trend" instead?
Re: Thanks for your comments. It has been corrected in the revised manuscript.
Some minor editing for adjustments of typos e.g. like 66 "(DHA )" - remove spacing and line 68 "[14,15].Moreover"- space required after full stop.
Re: Thanks for your comments. It has been corrected in the revised manuscript.
Table 2 footnotes "2the" T should be capatilzed
Re: Thanks for your comments. It has been corrected in the revised manuscript.
Reformatting of reference no. 13
Re: Thanks for your comments. It has been corrected in the revised manuscript.
Reviewer 2 Report
Dear Authors,
the manuscript ha been improved from the first version and all the comments has been adressed.
Author Response
Thanks for your comments.
Reviewer 4 Report
I regret to see that the authors do not satisfactorily reply point-by-point to my initial report. To check if they addressed my comments, I checked line by line because they did not explain in detail (reporting the new position of the revision) each comment that I suggested. Therefore, I reported again my comments.
The abstract is too long. According to the journal guidelines, the abstract should be about 250 words. The exposition can be improved.
At first glance, the abstract appears to be the same as the first submission. Then, I left the decision to the editors.
A simple summary section is missing. I suggest reducing the abstract and creating the Simple Summary section. In this section, the authors can explain and address the problems faced in their study.
Done. The authors addressed this point.
Line 55. Please report EFAs being for plural. Check this in the entire manuscript.
Before line 55, now line 63. Check if it is revised in the manuscript.
Line 58. EPA and DHA are not stated before abbreviating.
Before line 58, now line 66. Done.
Line 60. Explain the abbreviation CLA.
Before line 60, now line 69. Done.
In the introduction section the following sentence “Pigs were fed with the corn-soybean meal diet containing 2% SO or oil mixtures (FPRO and CPRO).” should be mentioned in the M&M section. The following statement “We observed an improved digestibility and intestinal epithelium functions, as well as a reduced production of inflammatory cytokines in the FPRO and CPRO group, which may offer a theoretical basis for rational selection of lipid sources for the weaned pigs as well as other mammalian animals.” should be stated in the conclusion.
This point has not been addressed by the authors. Please, check and revise the text according to the comments.
In the experimental design section, authors should indicate the duration of the study. This information is only mentioned in the abstract.
Done
Do not the authors believe the 28-day period is short? In my opinion, it should be a limitation of the study. The authors could create a short paragraph at the end of the manuscript explaining the limitations of the study and further perspectives. Alternatively, they could integrate this aspect into the conclusion.
This point has not been faced by the authors. If they believe that it is not necessary, they must provide a clear motivation.
Line 72. LCFA and MCFA should be reported in plural form then LCFAs and MCFAs. Check all abbreviations in the abbreviations section.
Done, but again check carefully in the text. It is not necessary to abbreviate a compound if it is no longer mentioned in the text.
Line 136. If an Abbreviation section is included in the manuscript, please use abbreviations.
This point should be addressed with the layout editor.
The letter P of the p-value should be reported as a lowercase letter.
Done
Figure 1. Regarding the effects of different dietary lipid sources on intestinal morphology, it would be interesting if the authors could illustrate an intestinal section of an individual who did not receive the dietary treatment to observe differences between a control subject and treated animals.
The authors have not provided any answer to address this point. An answer should be given. In my opinion, it could be useful to show an intestinal section of an individual that was not treated. It should allow us to compare a non-treated subject with the treated one to highlight differences.
I appreciate whether the authors could include in the bibliography of their manuscript the following research, highlighting in the discussion section the relevance of the use of alternative feedstuffs in pigs' nutrition for improving animal health, and growth performance.
Sutera, A.M.; Arfuso, F.; Tardiolo, G.; Riggio, V.; Fazio, F.; Aiese Cigliano, R.; Paytuví, A.; Piccione, G.; Zumbo, A. Effect of a Co-Feed Liquid Whey-Integrated Diet on Crossbred Pigs’ Fecal Microbiota. Animals 2023, 13, 1750. https://doi.org/10.3390/ani13111750
According to MDPI guidelines, the reviewer can encourage the author to cite studies to improve and update the bibliography of their manuscript. It seems that the authors did not include the suggested study in their manuscript.
Of course, it is not mandatory for the authors. Notwithstanding, it is polite to motivate if the authors decide to not follow the suggestion of including the reference in the bibliography.
English style can be improved.
Author Response
The abstract is too long. According to the journal guidelines, the abstract should be about 250 words. The exposition can be improved.
Re: Thanks for your comments. I have changed the abstract to an appropriate length.
Before line 55, now line 63. Check if it is revised in the manuscript.
Re: Thanks for your comments. It has been corrected in the revised manuscript.
In the introduction section the following sentence “Pigs were fed with the corn-soybean meal diet containing 2% SO or oil mixtures (FPRO and CPRO).” should be mentioned in the M&M section. The following statement “We observed an improved digestibility and intestinal epithelium functions, as well as a reduced production of inflammatory cytokines in the FPRO and CPRO group, which may offer a theoretical basis for rational selection of lipid sources for the weaned pigs as well as other mammalian animals.” should be stated in the conclusion.
Re: Thanks for your comments. It has been corrected in the revised manuscript.
Do not the authors believe the 28-day period is short? In my opinion, it should be a limitation of the study. The authors could create a short paragraph at the end of the manuscript explaining the limitations of the study and further perspectives. Alternatively, they could integrate this aspect into the conclusion.
Re: Thanks for your comments. It has been corrected in the revised manuscript.
Figure 1. Regarding the effects of different dietary lipid sources on intestinal morphology, it would be interesting if the authors could illustrate an intestinal section of an individual who did not receive the dietary treatment to observe differences between a control subject and treated animals.
Re: Thanks for your comments. We determined the effect of dietary lipids on intestinal morphology by making intestinal slices to read the villus height and crypt depth of the villus intestine. The manuscript has analyzed the effects of different lipids on the villus height and crypt depth of piglets, which can also reflect the impact of different dietary fibers on intestinal morphology.